# Analysis of the Effectiveness of Measures on the COVID-19 Vaccination Rate in Hong Kong

**DOI:** 10.3390/vaccines10050747

**Published:** 2022-05-10

**Authors:** Yui-Ki Chu, Pui-Hong Chung, Fei-Chau Pang

**Affiliations:** Li Ka Shing Faculty of Medicine, The University of Hong Kong, Hong Kong, China; chuyk@hku.hk (Y.-K.C.); fcpang@hku.hk (F.-C.P.)

**Keywords:** COVID-19 vaccine, vaccination measure, vaccination rate

## Abstract

Background: The World Health Organization has set a target of at least 70% of the global population being vaccinated by the middle of 2022. There are only 17 countries that achieved a 70% vaccination rate (VR). This study aims to analyze the effectiveness of public policies to increase the COVID-19 VR. Methods: vaccination figures of all eligible population groups in Hong Kong from 22 February 2021 to 23 January 2022, were extracted for analysis. Weekly acceleration in the VR (AVR) was calculated as a measure of policy effectiveness. A total of 13 identified measures were classified into four policy categories: eligibility, accessibility, incentives, and restrictions. Age-weighted AVR (AWAVR) was compared by age group and policy presence vs. absence using Mann–Whitney U tests. Results: the AWAVR means across age groups ranged from −1.26% to +0.23% (*p* = 0.12) for eligibility; accessibility ranged from +0.18% to +1.51% (*p* < 0.0001); incentives ranged from +0.11% to +0.68% (*p* < 0.0001); and restrictions ranged from +0.02% to +1.25% (*p* < 0.0001). Conclusions: policies targeting accessibility, incentives, and restrictions are effective at increasing the VR. These results may serve as a policy reference.

## 1. Introduction

The severe acute respiratory syndrome coronavirus 2 (SARS-CoV-2), publicly known as coronavirus disease 2019 (COVID-19), was declared a pandemic by the World Health Organization (WHO) on 11 March, 2020. As of 28 January 2022, there were 364,191,494 confirmed cases and 5,631,457 reported deaths globally [1]. Vaccines had been developed shortly to address the high transmission rate and high severity of the disease. The U.S. Food and Drug Administration had given temporary regulatory approval to the first COVID-19 vaccine, the Pfizer-BioNTech vaccine [2]. Vaccines are proven to reduce severe complications and death [3]. The WHO had set a target of at least 70% of the global population being vaccinated by the middle of 2022. However, there were only 17 countries that achieved a 70% vaccination rate (VR) as of 28 January 2022 [4]. Many countries and cities were struggling to raise the uptake rate among the rest of the world. 

A territory-wide vaccination program was introduced to the public by the Hong Kong (HK) government on 23 February 2021. Two vaccines were authorized, used in HK, (1) Fosun Pharma/BioNTech (BioNTech, Mainz, Germany), and (2) Sinovac Biotech (Hong Kong) Limited (Sinovac, Beijing, China), both were evident-proven to minimize possible complications [5,6,7]. The BioNTech vaccine was also found to lower the hospitalization rate against COVID-19 and reduce the chance of developing severe diseases [7,8].

For a population-based vaccination program, the VR of a region was driven by individuals’ perceptions of the risk of vaccination and the motivation to receive the vaccine. However, owing to cut-short approvals of the vaccines, the general population cast doubt on their safety and effectiveness [9]. The main hesitation of the local population was worrying about the safety, side effects, and effectiveness brought by the vaccine [9]. As of 10 January 2022, the 319th day since the implementation of the vaccination program, the VR in HK was 74.7% (admitted to the first dose) [10]. By comparing to our surrounding regions, HK’s VR was still lagging behind Singapore and China, which were 87% and 84%, respectively [11]. Nevertheless, the current rate had reached the 70% target rate set by the WHO. This study analyzes the impact of various measures on the VR for policy development that contribute to a successful vaccination program, attaining herd immunity. We conducted an analysis of the effectiveness of government- and corporate-implemented vaccination measures to estimate their efficacies in increasing VR in HK.

## 2. Materials and Methods

### 2.1. Study Design and Data Collection 

A quantitative analysis of the effectiveness of measures regarding the VR was conducted. Statistics that included all vaccination-eligible residents in HK between 22 February 2021 and 23 January 2022, were collected from the HK government’s daily reported vaccination statistics and daily reported confirmed cases website. The data were sorted by age groups, date of admission, and type of vaccine. The official press releases and news involving vaccination policies and measures in HK that were published during the above-mentioned period were reviewed to retrieve the government’s and corporates’ actions.

The first dose of the VR was used to analyze the COVID-19 VR trend, while it was analyzed separately by age groups. The age group classification followed the HK government vaccination dashboard, as 12–19, 20–29, 30–39, 40–49, 50–59, 60–69, 70–79, and 80 or above. All VRs were composed only of individuals who were eligible for vaccination.

### 2.2. Data Analysis

VRs of each age group were calculated by the first-dose injection number of the age group divided by eligible recipients reported in the HK government’s statistics. To visualize the association between the trend of the VR and the corresponding measures, differencing was applied in the VR, as a measure of the rate of change, to identify the boost in the VR toward the implemented policies. To smooth out the daily variance, a weekly resolution was adopted with end-of-week resampling. 

The effectiveness of each applied measure was evaluated by the second differencing in the weekly VR, as a measure of the VR acceleration (AVR), from the point of its adoption, such that the short-term VR-boosting effect of each policy was quantified and compared. To better evaluate the policy’s impact based on its targeted age group(s), age-weighted AVR (AWAVR) of the corresponding group(s) was calculated. Positive AWAVR indicated an acceleration in the VR of the targeted age group(s). They were calculated by
$${\mathrm{AVR}}_{t}={\mathrm{VR}}_{t}-2\times {\mathrm{VR}}_{t-1}+{\mathrm{VR}}_{t-2}$$
$$\mathrm{AWAVR}=\frac{\sum_{targetted\;age\;groups}\left(AVR_{age\;specific}\times age\;group\;population\right)}{Total\;population\;of\;targetted\;age\;groups}$$

Policies were classified into one of the four categories: eligibility, accessibility, incentive, and restriction. Tests of significance of each policy category were conducted, with the null hypothesis being the mean AWAVR changes of “policy-present” periods were the same as “policy-absent” periods. Four consecutive weeks since the week of policy interception was taken as the assumed policy effective period, in which that period would be considered as “policy-present”, while other timepoints as “policy-absent”. Each category’s policy presence was tested as the independent variable, whereas the related age groups’ AWAVR was the dependent variable. Shapiro–Wilk test was used for the normality check of the AWAVR. An independent *t*-test was applied if the AWAVR was normally distributed, while a Mann–Whitney U test was used if not. The significance level for all statistical tests was set at α = 0.05.

## 3. Results

### 3.1. Study Participants and Characteristics 

The data represented 6,734,600 of HK’s total eligible population for vaccines, aged over 12. A total of 10,790,013 doses of vaccines were given as of 23 January 2022. Among them, 61.7% (6,661,061/10,790,013) were BioNTech while 38.3% (4,128,952/10,790,013) were Sinovac. The total VR among the HK population was 77.9% (5,246,087/6,734,600). The distribution of vaccines is shown in Figure 1. Besides the 12–19 age group, juniors below 16 were only eligible for Sinovac. Younger age groups (20–29, 30–39) preferred BioNTech more than Sinovac while the older age groups preferred Sinovac more than BioNTech. 

### 3.2. Vaccination Rate by Age

The VR by age in percentage is plotted in Figure 2. The age group with the highest VR was 40–49 with 90.6%, followed by the age groups of 50–59, 30–39, 12–19, and 20–29 with 85.4%, 84.0%, 83.0%, and 82.8%, respectively. The older age groups had a lower VR of 72.8%, 57.5%, and 29.1% in the groups of 60–69, 70–79, and 80 or above, sequentially. 

### 3.3. Absolute Change in Weekly Vaccination Rate by Age 

There were three peaks in the change of the VR, as shown in Figure 3. The first peak appeared from 21 March to 18 April, 2021, with a sudden dropped in the VR in early April that might be due to the temporary suspension of the BioNTech vaccine. Weekly increments in the VR among age groups were up to 3.70%. The second peak was found from 13 June to 22 August, 2021, with a maximum of 5.50% (age 12–19), while the weekly increments in the VR of most age groups were above 3%. The last peak was observed in early January 2022. The weekly increments of VR raised from below 1% for all age groups to 1.19%–3.40%. 

The extent of the increment of change in the VR in the younger and middle age groups was found to be larger in the first and second peaks while the change in the VR of the elder age groups increased in a larger magnitude than the younger ones in the third peak. 

### 3.4. Effectiveness of Measures

Four categories of measures to increase the VR were classified into (1) eligibility, (2) accessibility, (3) incentive, and (4) restriction categories. The HK government, as well as some enterprises, launched a series of measures to reduce vaccination barriers or hesitancies and uplift the VR. These measures were target-oriented to those lower VR age groups. Our analysis aimed to study the impact of these measures that they had implemented (Table 1). 

#### 3.4.1. Eligibility

Eligibility mainly affected the VR of the minors as they were not eligible for vaccination at the early stage of the program since there was insufficient statistical data when the vaccine was first launched. The VR of the age group of 12–19 remained the lowest among age groups until late May 2021 (Figure 1). New evidence and the eligibility of the vaccines were constantly reviewed and evaluated by the HK government. On 15 April, 2021, it announced an extension of the eligibility of the BioNTech vaccine to those aged over 16 and Sinovac for teenagers aged over 18 [12]. The eligibility of Sinovac was further extended on 15 September 2021 to 12 years old or above. Yet, eligibility extension measures had no significant effect on the AWAVR of the 12–19 age group (*p* = 0.12).

#### 3.4.2. Accessibility 

Accessibility was comprised of how much effort an individual needed to get vaccinated. The numbers of CVCs, the ease of use of the vaccination booking system, and crowd management of the vaccination program were the main components. There were concerns about online booking difficulties for the elderly due to the lack of technological knowledge. Thus, two interventions were launched to reduce this vaccination barrier for the elderly: (1) outreach vaccination services, and (2) in-person same-day CVC vaccination quota distribution. The average post policies AWAVRs ranged from +0.18% to +0.37%. 

Guardians’ consent and accompany were required for minors aged 12–17 to have their vaccines. It increased the barrier to their accessibility to the vaccine. Group booking vaccination services and school outreach vaccination services were hence launched. The mean AWAVRs ranged from +1.16% to +1.51% post-policy interception. Providing convenience in vaccination to address the accessibility concern was found to be statistically significant (*p* < 0.0001).

#### 3.4.3. Incentive

Incentives measures, including extra authorized leaves and monetary incentives, were found to be statistically significant in the change of AWAVR (*p* < 0.0001). A series of monetary incentives were provided by corporations, including a free apartment worth USD 1.4 million, from 2 June 2021, for the HK residents who were vaccinated on or before 31 August 2021 [18]. The AWAVR of all eligible age groups increased to +0.68% (±0.56%). On the other hand, the AWAVR in the working-age groups (20–69) increased to +0.68% (±0.56%) when vaccination leaves were provided, while resuming face-to-face classes and school activities yielded +0.11% (±0.79%) in AWAVR for the 12–19 age group.

#### 3.4.4. Restriction

The effect of imposing restrictions on AWAVR was statistically significant (*p* < 0.0001). The effectiveness varied according to the tightness of the measures. The AWAVR of working-age groups was +0.11% (±0.79%) when the ‘once-in-every-two-week compulsory testing’ was introduced. It increased to +1.21% (±2.15%) when the compulsory testing frequency was increased to once every three days. The AWAVR surged to +1.25% (±2.12%) after the announcement of venue entry restriction without a vaccination record.

## 4. Discussion

The HK government managed the crowd by providing multiple CVCs in every district to facilitate the vaccination program. However, the VR of the elderly was still lagging behind the other age groups. The vaccination status of the minor and elderly in HK differed from that of the Western countries. In the UK, the younger the population, the lower the VR [24]. Similar vaccination statistics were found in the US—the older population tended to have a higher vaccination rate [25]. In HK, the VR for those aged 12–19 remained low in the beginning and raised after the accessibility intervention was introduced. The result of this study suggested that school outreach services might be effective in countries that struggle to increase the vaccination rate of teenagers. 

The lack of initiation could have resulted from the concerns about the safety, side effects, and effectiveness of the vaccines. A local study revealed that 76.4% of citizens worried about the side effects of the newly developed vaccines, whereas around 70% worried about their safety. Most admitted that observation of their effect was needed before making decisions [9]. The policies targeting the lack of initiatives were classified into incentives (reward) and restrictions (punishment) by using the operant conditioning theory to analyze the AWAVR [26]. Similar strategies were applied in other countries. A study in Sweden indicated that people had a higher chance of receiving the vaccine under the condition of offering monetary incentives [27]. A study on lottery influence conducted in Ohio, USA, also pointed out that there was a rise in the VR after the announcement of the Vax-a-Million lottery [28]. Relatively less research had discussed the effect of incentives with emphasis on age. Our study pinpointed that the incentive strategy was an effective measure to increase the VR applicable to all age groups, especially the working ones. Researchers had found that paid sick leave could increase the influenza VR [29]. This coheres with our research result, in which paid leave was lucrative to working-age groups for vaccination.

Self-financed testing and venue-entering limitations were introduced to the non-vaccinated population by the HK government. The VR accelerated along with the mandatory testing frequency. Work constraints brought to the working-age group induced inconvenience. Testing every two weeks was not contributing to the growth in the VR of the working-age group. The AWAVR of these age groups increased after the policy was tightened to once a week. The largest AWAVR increase was observed when the testing frequency became once every three days. The increase in compulsory testing was particularly effective for the working population to initiate their vaccination decision. Although a local study indicated that the vaccination acceptance rate ranked highest among individuals aged over 60, the VR of the age group above 60 remained the lowest in HK [9]. In this study, the overall VR drastically increased after the introduction of venue-entering restrictions. This increase was particularly significant among the population over 60. This showed that venue-entering restrictions greatly affect the habits of these retired people. Previous local research had indicated that entering entertainment facilities only had a low to medium effect on vaccination acceptance [30]. However, the VR vastly increased with the recent policy tightening. Thus, the effectiveness of the broader extent of venue-entering restriction suggested an increase in the VR for all age groups, especially the seniors. 

Apart from the measures that had been discussed, VR might also be affected by numerous confounding factors. (1) Confidence in vaccines: a study had proven the association between the decrease in vaccine willingness and its safety [31]. Serious adverse events, such as Bell’s palsy or death, were found to be a great barriers affecting vaccination intentions in HK [32]. This particularly affected the elders who were concerned about being unsuitable for vaccination due to age and medical history that might lead to a higher risk of induced death [33]. The general population would balance the safety and effectiveness of the newly developed vaccine for their vaccination intension and choice. The elderly tended to stress the side effects of the vaccines. This study found that older age groups preferred Sinovac, in which media claimed that it had less severe overall side effects. Compared to the younger generation, they might be more reliant on scientific evidence, or the overall effectiveness, such that they might be prone to select BioNTech; thus, proving that more vaccine choices could suit different population needs and raise the VR. (2) Media: rumors and anti-vaxxer campaigns expressed their untrust towards the vaccines or the HK government might confuse the public. Propaganda could highly affect the confidence in vaccinations due to the unknown of the newly emerged virus. There was an association between confidence in vaccines and health literacy [34]. The HK government constantly clarified the misinformation through daily press conferences and promoted proper health education through TV advertisements throughout the pandemic. However, due to the constant extent of advertising, its effectiveness was difficult to be quantified and compared. (3) Infection figure: US research has indicated a negative association between the VR and infection rate [35]. This was also reflected in HK before the end of 2021. However, there was a surge in both the VR and the number of confirmed cases since the end of 2021. Therefore, the possibility of the number of confirmed cases being a facilitator of vaccination cannot be neglected. The mentioned factors might serve as vaccination hesitancies or boosters; therefore, policymakers should put these into account during vaccination policymaking.

Several limitations have been acknowledged. As some of the policies were announced concurrently, the AWAVR change could result from more than one factor. The effectiveness of the introduced policy might be over-estimated since the AWAVR in the target group might be subjected to the collinearity effect, while the extent of the co-effects of multiple policies was hard to be quantified. The efficiency of the measures was not taken into account; hence the assumed universalized effective duration of 4 weeks might not accurately reflect the specific effectiveness. To further investigate the influencing factor of the increase in the VR, an in-depth survey should be conducted.

## 5. Conclusions

The world is now encountering an unprecedented pandemic. Receiving the vaccine is seemingly a possible solution to this crisis. To achieve so, a clear strategy should be set, particularly applied to those countries with low vaccination compliance, with consideration of their demographic structures. Four types of vaccination measures were sorted by this study—eligibility, accessibility, initiatives, and restrictions. Outreach vaccination services and booking free, same-day vaccination ticket distributions, were found to be effective to minimize the accessibility barrier among students and the elderly. To enhance the initiative, providing incentives, such as lottery and vaccination leave, was found to be a potential strategy to increase the VR among the working group. Broad restrictions on venue-entering, with the inclusion of restaurants and public facilities, posed a high efficacy to boost the VR for all populations. The high VR rate in HK was contributed by the high degree of accessibility and provided initiatives. The research findings could apply to vaccination programs in other countries and future pandemics.

## Figures and Tables

**Figure 1 vaccines-10-00747-f001:**
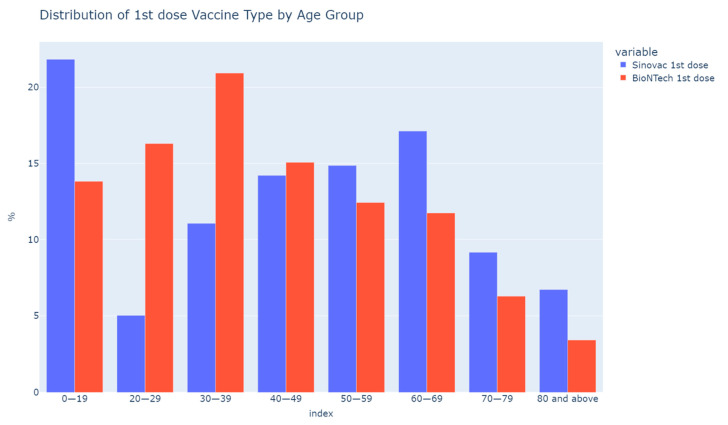
Distribution of first dose vaccine type by age group.

**Figure 2 vaccines-10-00747-f002:**
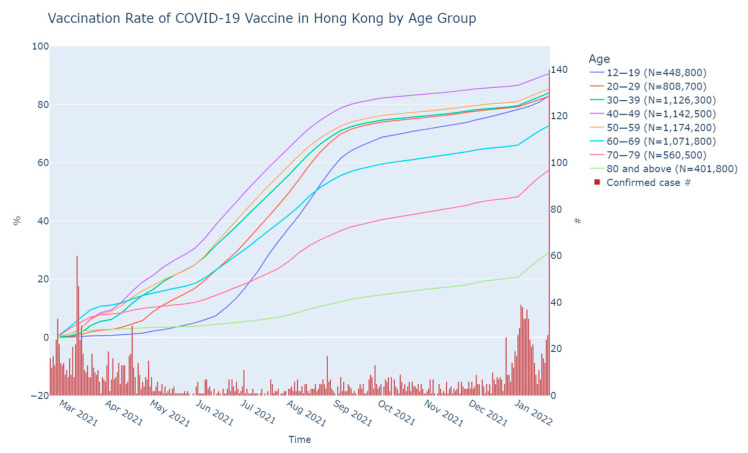
Vaccination rate of COVID-19 vaccination in Hong Kong by age group.

**Figure 3 vaccines-10-00747-f003:**
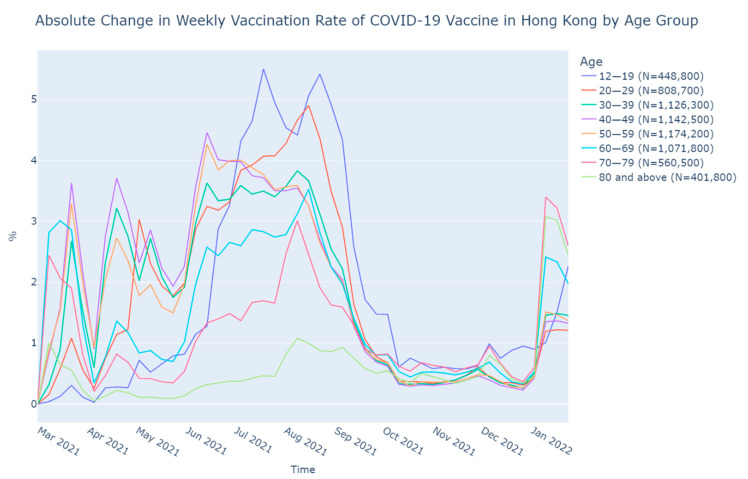
Absolute change in weekly vaccination rate of COVID-19 vaccine in Hong Kong by age group.

**Table 1 vaccines-10-00747-t001:** Effectiveness of measures.

Category of Measure	*p*-Value ^1^	List of Measures	Date of Announcement	Target Age Group	AWAVR Mean (SD) ^2^
Eligibility	0.12	Extended eligibility of BioNTech and Sinovac to those aged 16 or above and 18, respectively [12]	15 April 2021	12–19	+0.23% (0.35%)
		Extended eligibility of Sinovac to 12–17 [13]	15 September 2021	12–19	−1.46% (1.29%)
Accessibility	<0.0001	Launched group booking vaccination services [14]	21 June 2021	12–19	+1.51% (0.99%)
		Provided outreach vaccination arrangements for Residential care homes [15]	13 April 2021	60 or above	+0.18% (0.63%)
		Launched government’s school outreach service for COVID-19 vaccination [16]	2 July 2021	12–19	+1.16% (0.59%)
		Distributed in-person same day quota that no need booking for elderly who aged 70 or above [17]	29 July 2021	70 or above	+0.37% (0.39%)
Incentives	<0.0001	Provided vaccination leave for government employees who injected first dose on or before 31 August, 2021. Other companies followed suit [16]	31 May 2021	20–69	+0.68% (0.56%)
		Launched lucky draw by Sino Group. Over 30 enterprises followed suits [18]	2 June 2021	18 or above	+0.68% (0.56%)
		Announced resume face-to-face whole day classes and school activities when 70% of students are vaccinated [19]	2 August 2021	12–19	+0.11% (0.79%)
Restrictions	<0.0001	Demanded all government and hospital authority staff without vaccination to undergo USD 30 self-paid combined nasal and throat swabs once every two weeks outside their working time [20]	2 August 2021	20–69	+0.11% (0.79%)
		Further tightened the testing scheme for those who had not been vaccinated from once in two weeks to once a week [21]	9 November 2021	20–69	+0.02% (0.10%)
		Further tightened testing schedule from once a week to every three days [22]	21 December 2021	20–69	+1.21% (2.15%)
		Forbidden entering all venues without the proof of vaccination certificate [23]	4 January 2022	All	+1.25% (2.12%)

^1^ Mann–Whitney U test. ^2^ 4 weeks after the announcement date.

## Data Availability

The data used and analyzed during the current study are available from the corresponding author upon reasonable request.

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
