# Peer review of "Analysis of the Effectiveness of Measures on the COVID-19 Vaccination Rate in Hong Kong"

_vaccines, 2022, doi:10.3390/vaccines10050747_

Round 1
Reviewer 1 Report
Sentance structure and grammatical errors make paper difficult to read in areas. Editing by a native English speaker in consultation with the authors to clarify meaning where necessary would improve readability of the paper.
Use of acronyms does not help readability. I could not always easily find first use to check for meaning. A table of "acronyms" used would help in this regard.
Age group 0-19 comprises a majority of subjects ineligible to be vaccinated. It is not clear to me that age-weighting compensated for this. Please clarify. Change in numbers of people vaccinated as well as percent changes would be helpful. As would demographics of the Hong Kong population by age group.
I appreciate the authors' acknowledgement of colinearity and potential shared varience as a limitation of the study. A further discussion of other possible confounding factors would be appreciated.
Choice of vaccine--as data on vaccine effectiveness was released this seemed to affect patient confidence in one or other of the available vaccines which may, in turn, have affected vaccination rates, e.g., if my confidence in the only vaccine available to me is low I may be less likely to become vaccinated. Consider also differental risk, i.e., vaccines with greater documented risk (e.g., blood clots) may be less appealing than those perceived as lower risk.
Infection rates, hospitalizations, mortality--changes in these data and corresponding changes in perception of risk may also have affected vaccination rates. An overlay of changing infection rates against changes in vaccination rate would be interesting to see.
"Advertising"--were there any media campaigns aimed at increasing vaccination rates in any general or specific populations running at the same time as the researched interventions?
Anti--vaxxers--populations with more active anti-vaccination campaigns may have lower vaccination rates.
Author Response
Thank you very much for your time to review our manuscript. We have made amendments according to your comments. Please review, thank you.
An abbreviation table has been added to Appendix I for your reference.
The age group 0-19 have been changed to 12-19 due to the vaccine eligibility as of 23 January 2022 in Hong Kong.
If we present the change in the numbers of people vaccinated would be like the graph as follows. However, it would be visually messy with the same interpretation. It is not as useful as a change in VR% as a reference for policymaking.
Link to image
In the discussion part, three confounding factors were further discussed.
(1) confident in the vaccine (different types of vaccine)
- safety and effectiveness concerns of the public
- age and medical history concerns
- the risk of possible or serious adverse events
(2) media
- rumours about vaccines and anti-vaxxer campaigns would affect people’s intention on taking vaccines.
(3) infectious figure
- Infectious figures as of 23 January 2022 were added to Fig.2 compared with the vaccination rate in Hong Kong.
For hospitalizations and mortality, there were no regular data on hospitalizations and mortality published by the government as of 23 Jan 2022.
For advertising and the Anti-vaxxers population problem, I added points in the discussion to address your concerns.
Reviewer 2 Report
- the authors should discuss the role of forcing vaccination by Health Institution on the overall VR. Please discuss such a point.
- the authors should evaluate the impact of education on the final results. The impact of education degree might effectively influence the overall VR of the vaccines.
- what about the impact of web information and fake news on vaccines? Please discuss such a point.
- A capillary organization of the hubs of the vaccines and the management of the crowds and population is fundamental for the VR success. Please discuss such a point.
- the alarmistic news from the media might impact on the awareness of the population on the need for vaccine. Please discuss this further point.
Author Response
Thank you very much for your time to review our manuscript. We have made amendments according to your comments. Please review, thank you.
For the first point, there was no forcing vaccination in Hong Kong but the government highly encourage the public by providing incentives and restrictions.
For the second and third points, we added discussion on the perceptions and health knowledge on the vaccines and confounding factors including fake news and anti-vaxxer promotion.
For the fourth point, we agreed the capillary organization of hubs and management of crowds are a crux to success and we have added a concluding line to the conclusion. We have pointed out it could be a colinear effect which made the extent of the co-effects of multiple policies hard to quantify.
For the last point, we have added points in the discussion to further discuss this point.
Thank you very much!
Round 2
Reviewer 1 Report
Editing for English language and style is minimal in this revision. The paper is still difficult to read, including the updated additions.
Edited example for Abstract:
Weekly acceleration in VR (AVR) was calculated as a measure of policy effectiveness. A total of 13 identified measures were classified into 4 policy categories: eligibility, accessibility, incentives, and restrictions. Age-weighted AVR (AWAVR) was compared by age group and policy presence vs absence using Mann-Whitney U tests (p=0.05). Results: The AWAVR means across age groups ranged from -1.26% to +0.23% (p=0.12) for eligibility; accessibility ranged from +0.18% to +1.51% (p<0.0001); incentives ranged from +0.11% to +0.68% (p<0.0001); and restrictions ranged from +0.02% to +1.25% (p<0.0001). Conclusions: Policies targeting incentives and restrictions are effective in increasing the VR. These results may serve as a policy reference.
I would suggest to the authors one of their cited papers (ref # 33) as a model for English usage and readability.
Figures are "blurry" and difficult to read.
I would still like information on numbers in each age group. This could be added to Fig 1. as "N=" under each age group without being too "visually messy".
Author Response
Thank you for your comment and suggestions.
We have taken Ref 33 as a reference for the English writing style and rewritten the manuscript.
Figures were replaced with a high-resolution version and added (N) next to each age group.
Thank you very much!
Reviewer 2 Report
the authors well addressed my previous comments. THe paper improved very much
Author Response
Thank you very much!
Round 3
Reviewer 1 Report
Readability is much improved. There are still a number of grammatical errors, and English usage and word choice issues that would benefit from further editing.